# DipDNN - Decomposed Invertible Pathway Deep Neural Networks

## Abstract

Deep neural networks (DNNs) enable highly accurate one-way inferences from inputs to outputs. However, there's an elevated need for consistency in bi-directional inferences, such as state estimation, signal recovery, privacy preservation, and reasoning. Since standard DNNs are not inherently invertible, previous works use multiple DNNs in a nested manner to obtain consistent and analytical forms of inverse solutions. However, such a design is not only computationally expensive due to DNN compositions, but also forces splitting the input/output equally, which is inapplicable in many applications. To reduce the restriction, other works use fixed-point iterations to enable approximation of one-to-one mapping, but the numerical approximation leads to reconstruction errors compared with the analytical inverse. To preserve the analytical form with minimum computational redundancy, we proposed decomposed-invertible-pathway DNNs (DipDNN) that decompose the nested design. We enforce one-to-one mapping in each layer by minimally adjusting the weights and activation functions of standard dense DNNs. We prove that such an adjustment guarantees strict invertibility without hurting the universal approximation. As our design relaxes the alternative stacking of nested DNNs, the proposed method does not need a fixed splitting of inputs/outputs, making it applicable for general inverse problems. To boost the two-way learning accuracy further, we show that the proposed DipDNN is easily integrated into a parallel structure. With the analytical invertibility, bi-Lipschitz stability regularization naturally fits into the scheme to avoid numerical issues. Numerical results show that DipDNN can recover the input exactly and quickly in diverse systems.

## 1 Introduction

Deep neural networks have shown success in various examples to make inferences accurately (LeCun et al., 2015; Schmidhuber, 2015). However, the high accuracy in one-way mapping is insufficient to fulfill diversified needs (Raissi et al., 2019; Kamyab et al., 2022). For many deterministic systems, especially physical systems, the complete modeling is bi-directional and covers both forward and inverse mappings for inferences (Tarantola, 2005; Bu & Karpatne, 2021). For example, recovering audio/image signals from data is the inverse process of regular transformations (Arridge et al., 2019). For most physical and engineering systems, estimating the hidden states/parameters is based on the forward system identification (Jensen et al., 1999). Therefore, the topics are popular in recent years in retaining sensitive information in privacy-preserving models and providing explanations for black-box models (Berman et al., 2019; Mahendran & Vedaldi, 2015). Both necessitate tracing back the decision-making process to its origin.

However, making the forward and inverse DNN mappings compatible is difficult. The challenges come from the multi-layered nonlinear structure and complex interconnections within layers of DNN, which do not naturally have one-to-one correspondence. To avoid these issues, previous methods have two major directions: either building a nested structure to avoid dealing with undesirable many-to-one property from DNN or constraining DNN parameters for a contractive mapping numerically (Dinh et al., 2014; Behrmann et al., 2019). Specifically, the nested structure requires a fixed splitting of input and output dimensions to retain DNN nonlinearity in an analytically invertible model. However, it raises problems with unfairly grouped data dimensions and the increased computational burden of separate DNNs. For example, the heuristic grouping of variables in physical system creates inconsistency to the physical structures. On the other hand, numerical approximation methods,

such as i-ResNet, relax the restrictive architecture with the cost of analytical inverse form. The reconstruction error is thus unavoidable, e.g., dependent on the convergence and accuracy of the fix-point iterations for inverse computation.

Therefore, is it possible to preserve the analytical inverse solution for applications that need accurate point estimates while reducing the computation redundancy of previous methods? For such a question, we firstly show how to convert the invertible structure with nested DNNs into a regular DNN with Leaky ReLU activation functions with performance guarantees on both the forward mapping and inverse learning. Such a design is based on the trade-off between the strict analytical invertibility and the model's approximation efficiency during the conversion. Motivated by that, we finalize our design with a decomposed invertible pathway DNN (DipDNN) model (Fig. 2). DipDNN minimizes the model redundancy/sparsity without hurting the approximation capability while ensuring an easy-to-compute inverse solution. Moreover, our proposed method relaxes the restrictions on invertible architecture, which does not require splitting input/output data or alternatively concatenating several DNNs (i.e., at least three and normally uses four for full-dimension couplings). Such properties greatly widened the application fields for inverse learning.

In addition to the analytical one-to-one correspondence, we introduce regularization on both forward and inverse processes to boost performance. For many vision-related works, the inverse problem has been formed as estimating the density of complex distribution, for which the generative learning models can have poor generalizability for data beyond training ranges (Nalisnick et al., 2019; Fetaya et al., 2020). To improve the extrapolation capability of the forward approximation, we introduce a trusted physics expert to compete and collaborate with the DipDNN and find the optimal split in function approximation. Although we have theoretical guarantees on the invertibility of DipDNN, numerical errors are not rare when coding in practice. As there is a trade-off between enforcing numerical inverse stability and maintaining approximation capability (Bal, 2012; Gottschling et al., 2020), we propose to find the balance with moderate regularization (Amos et al., 2017), which is shown to be both effective and robust in experiments. The numerical validation conducted on a variety of systems assesses forward accuracy, computational efficiency, and inverse consistency. The competitive performance shows that a basic adjustment of network layers can dramatically widen the application fields for DNN with bi-directional information flow.

## 2 INVERSE PROBLEM AND THE INTRICACIES

### 2.1 DEFINE THE GENERAL INVERSE PROBLEM

System identification is a supervised learning task to recover the forward mapping $f : \mathbb{R}^n \to \mathbb{R}^n$ with $\boldsymbol{y} = f(\boldsymbol{x})$ from measurements. There are cases when one need to know the hidden states or original variables. The inference function to know $\boldsymbol{x}$ is in reverse to function $f$. And, this paper focuses on a deterministic setup to obtain accurate point estimates of $\boldsymbol{x}$, which is different from the generative density estimation task via maximizing likelihood (Dinh et al., 2014). We aim to find an inverse mapping $g : \mathcal{Y} \to \mathcal{X}$ corresponding with $f$, which satisfies $\boldsymbol{x} = g(\boldsymbol{y}) = f^{-1}(\boldsymbol{y}), \forall \boldsymbol{y} \in \mathcal{Y}$. Unlike forward mapping, which usually has a well-defined governing function, the inverse counterpart is much more complicated to analyze (Bal, 2012). It can have multiple solutions due to ill-posedness. Unifying the learning of bi-directional mappings can address the issues via a consistent forward-inverse function pair. The learning task is to let $\boldsymbol{y} = g^{-1}(\boldsymbol{x})$ approximate the analytical forward model $\boldsymbol{y} = f(\boldsymbol{x})$ using historical data $\{\boldsymbol{x}_i, \boldsymbol{y}_i\}_{i=1}^N$, where the empirical errors are minimized. And, the approximation model $g^{-1}(\cdot)$ is designed to be an invertible structure. After the forward mapping is well-trained, the inverse mapping $\boldsymbol{x} = g(\boldsymbol{y})$ is obtained. It is expected to be explicitly consistent with the forward counterpart, minimizing the reconstruction loss of $g^* = \mathrm{argmin}_{g \in \}} \sum_{i=1}^N \ell_2 (\boldsymbol{x}_i, g^{-1}(g(\boldsymbol{x}_i)))$.

### 2.2 REVIEW ANALYTICAL INVERTIBLE TRANSFORMATION IN ADDICTIVE COUPLING LAYERS

Instead of recovering a direct inverse NN mapping, the unified learning strategy needs accurate approximation, invertibility of the forward NN, and an easy inverse computation. In literature, the invertibility can be enforced by normalizing the Lipschitz condition for an overall contractive DNN mapping and stable inverse, but the non-analytical inverse form can rarely

reach zero error during training for a perfect match of the two-way mappings. On the contrary, an analytical inverse requires DNN reconstruction to build one-to-one correspondence. A typical method of the analytical inverse is using the addictive coupling for invertible transformation (Dinh et al., 2014). For each layer, the invertibility is enforced by a fixed split of inputs $\boldsymbol{x} = [\boldsymbol{x}_{I_1}, \boldsymbol{x}_{I_2}]$ and outputs $\boldsymbol{y} = [\boldsymbol{y}_{I_1}, \boldsymbol{y}_{I_2}]$,

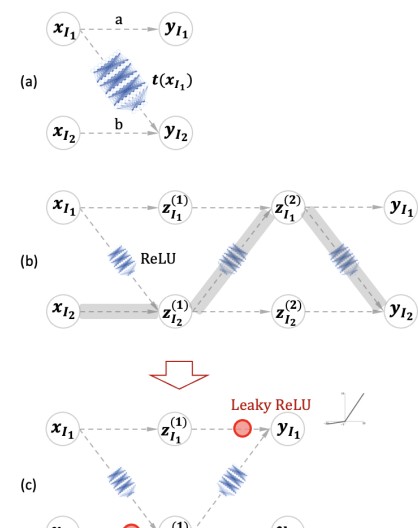

$$\boldsymbol{y}_{I_1} = a\boldsymbol{x}_{I_1}, \boldsymbol{y}_{I_2} = b\boldsymbol{x}_{I_2} + t(\boldsymbol{x}_{I_1}); \qquad (1)$$
$$\boldsymbol{x}_{I_1} = \boldsymbol{y}_{I_1}/a, \boldsymbol{x}_{I_2} = (\boldsymbol{y}_{I_2} - t(\boldsymbol{x}_{I_1}))/b. \qquad (2)$$

As shown in Fig. 1(a), $t(\cdot)$ can be arbitrarily complex, and we assign an MLP with ReLU activation functions. Invertibility requires the real function to have a one-to-one correspondence ($x_1 \neq x_2 \Rightarrow f(x_1) \neq f(x_2)$) of the inputs and outputs. Although $t(\cdot)$ creates a many-to-one mapping with ReLU, Fig. 1(a) uses a nested structure to build in the DNN without violating the overall invertibility. Specifically, the nested design requires splitting inputs and outputs, and the nonlinear coupling is limited between $\boldsymbol{x}_{I_1}$ and $\boldsymbol{y}_{I_2}$ within one layer of transformation. The invertibility is thus intuitive in that $\boldsymbol{x}_{I_1}$ and $\boldsymbol{y}_{I_1}$, $\boldsymbol{x}_{I_2}$ and $\boldsymbol{y}_{I_2}$ have one-to-one correspondence through the linear paths $a$ and $b$ (identical in the original design). And, the coupling between $\boldsymbol{x}_{I_2}$ and $\boldsymbol{y}_{I_1}$ is eliminated to easily derive the analytical inverse in equation 2.

Figure 1: (a) - (b): Composition of addictive invertible transformation for full coupling of input/output dimensions. (c): A reduction of (b) that retains full dimension coupling.

Therefore, to mimic the fully connected DNN, e.g., any output can have a nonlinear correlation with any input, the nested design needs to concatenate at least three layers alternatively. Mathematically, the necessity of concatenating three or more layers is intuitive by deriving the Jacobian matrix (derivation in Appendix A.1). Assuming we use the same architecture (i.e., $K$-layer ReLU-activated DNNs) for the nonlinear function in each layer, Fig. 1(b) needs three such DNNs to complete the coupling among all dimensions. The separate DNNs shall weaken the approximation capability so that four or more layers are usually concatenated in implementation. It could also aggravate error propagation in the inverse computation, which we will discuss later in Sec. 4.2.

# 3 ENFORCE INVERSE CONSISTENCY AND APPROXIMATION EFFICIENCY IN NEURAL NETWORK ARCHITECTURES

## 3.1 CAN WE REDUCE COMPUTATIONAL TIME?

The addictive coupling layer provides an analytical inverse, but it requires at least three layers to concatenate in turn for full nonlinear couplings of all the inputs with all the outputs, which normally takes four or more. Can we reduce the computational time with a flexible invertible structure?

By observing Fig. 1(b) and the Jacobian matrix, the redundancy of the nested design comes from the asymmetric coupling in a Z-shape. The sole purpose of the third stacking layer is to build a nonlinear mapping from $\boldsymbol{X}_{I_2}$ to $\boldsymbol{y}_{I_2}$. Can we reduce the layer for a lower computation burden? To shorten three compositions to two, we need to let $\boldsymbol{x}_{I_2}$ in the first layer contribute nonlinearity to the output at the bottom while reserving the invertibility. The one-to-one correspondence can be maintained by adding a Leaky ReLU activation function to the second path, which allows Fig. 1(b) to turn into Fig. 1(c). Based on the intuitive equation 1, we only change the direct path from linear correlation to a strictly monotonic nonlinear mapping for each layer, thus preserving the invertibility.

The structure in Fig. 1(c) still needs a hard division of inputs/outputs. Although the nonlinear DNN is nested in the middle, some interconnections among variables are eliminated due to the separated input/output groups, for which the comparison with regular NN is in Appendix A.1. Thus, it could be heuristic to find the optimal split for variables, e.g., measurements with physical meanings.

### 3.2 PROPOSED DECOMPOSED INVERTIBLE PATHWAY DEEP NEURAL NETWORKS

Previous methods have the typical splitting design to obtain an easy-to-compute Jacobian determinant, e.g., all 1's, for maximizing likelihood of training the generative model with unlabeled data. Since we target accurate point estimates rather than density estimation, the sparse structure is not necessary. We aim to construct one-to-one correspondence with minimum adjustment in DNN to maintain the dense function representation for universal approximation.

**Constructing Invertible DipDNN.** To keep the dense representation in a regular neural network layer: $\boldsymbol{z} = g(W\boldsymbol{x} + \boldsymbol{b})$, only two design choices are available for DNN layers: 1) activation function and 2) weights. For the nonlinearity $g(\cdot)$, the activation is element-wise such that strict monotonicity is a necessary and sufficient condition for a one-dimensional function to be invertible. We propose to use Leaky Rectified Linear Unit (Leaky ReLU) activation which is a strictly monotone function customized from ReLU.

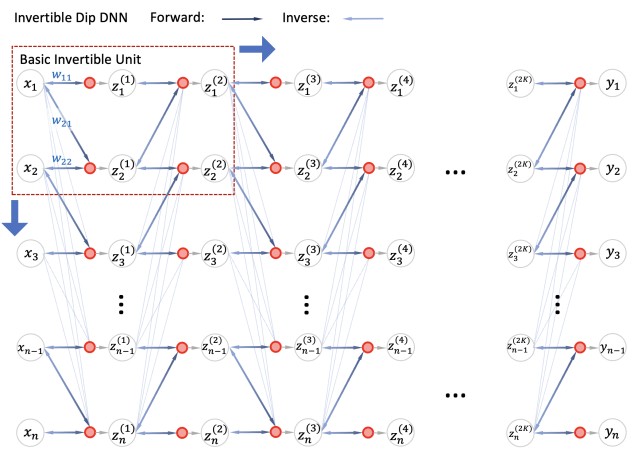

Figure 2: The proposed invertible DipDNN.

To make the affine function $W\boldsymbol{x} + \boldsymbol{b}$ bijective, the weight matrix $W$ needs to be invertible. It indicates independent correlation over all dimensions, where the one-to-one (injective) mapping means full column rank of the matrix and onto (surjective) means full row rank of the matrix. A non-singular square matrix always satisfies such one-to-one correspondence, but singularity issues may exist and cause difficulty in inverse computation. Motivated by the triangular map, a basic invertible unit (shown in the top right corner of Fig. 2) can eliminate the issue (details in Appendix A.2). As an extension of the basic invertible unit in depth and width, we propose lower and upper triangular weight matrices to render layers invertible. It can be seen as an equivalent adjustment using LU decomposition if we let $g_1$ be linear, i.e., $W = W_{tril}W_{triu}$ and the easily computed matrix inverse $W^{-1} = W_{triu}^{-1}W_{tril}^{-1}$ layer-by-layer. While any triangular matrix is invertible, if and only if all the entries on its main diagonal are non-zero, we alternately enforce the lower and upper triangular weight matrices in each block equation 3 to ensure the complete coupling over all the dimensions.

Therefore, Fig. 2 presents our compact invertible DNN structure. Mathematically, the proposed model has $K$ blocks, which indicate $2K$ layers. The representation for the $k^{th}$ block is:

$$\boldsymbol{z}^{(2k-1)} = g_1(W_{tril}^k \boldsymbol{z}^{\left(2(k-1)\right)} + \boldsymbol{b}_1^k), \boldsymbol{z}^{(2k)} = g_2(W_{triu}^k \boldsymbol{z}^{(2k-1)} + \boldsymbol{b}_2^k). \tag{3}$$

Each block $h^{(k)}$ consists of two layers $\boldsymbol{z}^{(2k-1)}$ and $\boldsymbol{z}^{(2k)}$. The model parameters include weight matrices $W_{tril}^k, W_{triu}^k$ and bias $\boldsymbol{b}_1^k, \boldsymbol{b}_2^k$. $g_1, g_2$ are element-wise nonlinear activation functions, and we use Leaky ReLU activation with a fixed parameter $\alpha \in \mathbb{R}^+ \setminus \{1\}, g(x) = \sigma_\alpha(x) = \begin{cases} x, & \text{if } x > 0, \\ \alpha x, & \text{if } x \leq 0, \end{cases}$.

The invertibility of the DNN model constructed in equation 3 is summarized in the following.

**Proposition 1.** *The function of the neural network model* $h : \mathbb{R}^n \to \mathbb{R}^n$ *with* $h = h^{(1)} \circ \cdots \circ h^{(K)}$ *is invertible if the weight matrices* $W_{tril}^k, W_{triu}^k, k \in [1, K]$ *are lower and upper triangular matrices with non-zero diagonal components, and all the activation functions* $g_k^1, g_k^2$ *are strictly monotonic.*

As the proposed model is a deep neural network structure with Decomposed Invertible Pathways layer-by-layer, we call it DipDNN, where "dip" also stands for the lower and raised connections.

**Preserving Representation Power.** Compared with Fig. 1, DipDNN relaxes the fixed input/output dimension splitting, thus no need to stack multiple separate DNNs alternatively for full couplings among groups. Meanwhile, instead of arbitrary nested DNN, DipDNN enforces the number of

neurons in all the layers to be the same for strict one-to-one correspondence. Will this weaken the representation power?

The universal approximation property of shallow wide networks (fixed depth such as one hidden layer and arbitrary width) has been well-studied, but it is still enduring work for deep narrow networks (bounded width and arbitrary depth). Especially, our DipDNN is a deep narrow network with weight matrices being alternatively lower and upper triangular. Next, we present the preserved universal approximation property in

**Theorem 1.** *Let $\mathcal{K} \subset \mathbb{R}^{dx}$ be a compact set, then for any continuous function $f \in C(\mathcal{K}, \mathbb{R}^{dy})$, there is a positive constant $\epsilon > 0$, $\|h(x) - f(x)\| < \epsilon$, for neural network $h : \mathbb{R}^{dx} \to \mathbb{R}^{dy}$, where $h = h^{(1)} \circ \cdots \circ h^{(K)}$. $h^{(k)}$ is defined in equation 3 with Leaky ReLU activation function and alternative lower and upper triangular matrices, $W_{tril}$ for odd layers and $W_{triu}$ for even layers.*

*Proof.* To describe the universal approximation of DNNs, we say the DNNs $h$ are dense in $C(\mathcal{X}, \mathcal{Y})$, if for any continuous function $f \in C(\mathcal{X}, \mathcal{Y})$, there is $\epsilon > 0$, such that $\|h(x) - f(x)\| < \epsilon$. To prove the universal approximation of DipDNN, we first refer to the latest results on the deep narrow Networks with Leaky ReLU activations as follows (Duan et al., 2023).

**Theorem 2.** *Let $\mathcal{K} \subset \mathbb{R}^{dx}$ be compact. Then the set of Leaky ReLU-activated neural networks with fixed width $d + 1$ ($d_x = d_y = d$) and arbitrary depth is dense in $C(\mathcal{K}, \mathbb{R}^{dy}) C(\Omega, \mathbb{R}^m)$.*

Theorem 2 indicates that there exists a neural network $h_\phi$ of lower bounded width $d + 1$ such that $\|h_\phi(x) - f(x)\| < \epsilon/2$. To convert $h_\phi$ to networks with triangular weight matrices, we denote the layer as $h_\phi(x)^k = \sigma(W^k h_\phi(x)^{(k-1)} + b^k)$. Since the dimensions in all layers are equal, each square matrix $W^k, k = 1, \cdot, K$ can be decomposed into a product of lower and upper triangular matrices, $W^k = W_{tril}^k W_{triu}^k$. The layer function turns to $h_\phi(x)^k = \sigma(W_{tril}^k W_{triu}^k h_\phi(x)^{(k-1)} + b^k)$. Subsequently, we transfer $h_\phi(x)^k$ into two layers by first inserting an identity map $I : \mathbb{R}^d \to \mathbb{R}^d$ and obtain $h_\phi(x)^k = \sigma(W_{tril}^k I W_{triu}^k h_\phi(x)^{(k-1)} + b^k)$. Then we apply some function $\rho^k : \mathbb{R}^d \to \mathbb{R}^d$ to approximate $I$ with $h_\psi(x)^k = \sigma(W_{tril}^k \rho^k W_{triu}^k h_\phi(x)^{(k-1)} + b^k)$. From the theorem on the identity mapping approximation (Liu et al., 2022), we construct $\rho^k$ to obtain $h_\psi(x)^k = \sigma(W_{tril}^{k'} \sigma(W_{triu}^{k'} h_\phi(x)^{(k-1)} + b^{k'}) + b^k)$, where $W_{tril}^{k'}, W_{triu}^{k'}$ are scaled by $\rho^k$, with structures remaining to be lower/upper triangular matrices. The approximation of identity mapping can reach arbitrary accuracy, and thus we have $\|h_\phi(x) - h_\psi(x)\| \leq \epsilon/2$. Given that $\|h_\phi(x) - f(x)\| < \epsilon/2$, we obtain $\|h_\psi(x) - f(x)\| < \epsilon$. Details of Theorem 2 and the theorem on the identity mapping approximation are included in Appendix A.2. $\square$

The result shows that any continuous function $f : \mathbb{R}^{dx} \to \mathbb{R}^{dy}$ can be approximated. To fit perfectly, DipDNN needs a slight construction to only expand the input and output dimensions from $d_x, d_y$ to $d + 1$ by filling in zeros without changing the property (Zhang et al., 2020).

## 4 REGULARIZATION FOR BOOSTING PERFORMANCE IN DIPDNN

While the conditions mentioned earlier guarantee analytical invertibility for a consistent inverse, the computation aspects of deep learning may raise a lack of generalizability and numerical stability issues, as supported by empirical observations. Recent analyses further provide theoretical evidence for the trade-off between approximation accuracy and inverse stability. In the following, we demonstrate the regularization scheme to train DipDNN and compute the inverse.

### 4.1 PHYSICS EMBEDDING

Common to discriminative learning tasks, the forward learning process's objective is to minimize empirical errors, a goal that hinges on the model's approximation capability. However, universal approximators can have excellent performance on training data but significant deterioration on out-of-distribution data. The unpredictable generalization is critical for physical systems that have changing operation points.

For many cases, such as physical systems, the forward model $f$ has specific priors or exhibits specific properties. Recent works on physics-informed learning embed these properties into the DNNs to

improve the generalizability. However, when it comes to the inverse problem, directly adding the symbolic embedding or extra constraints can break the invertibility of the forward mapping. Moreover, the exact underlying function may not naturally satisfy one-to-one mapping, and the inverse learning is only to approximate partially based on the observed data, which may cause conflict.

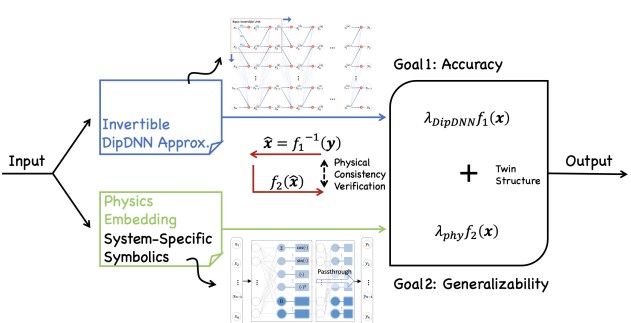

Figure 3: A parallel structure for physical regularization over DipDNN.

Therefore, we propose a twin structure in Fig. 12. A physics embedding is added in parallel with the DipDNN instead of embedding it into the forward model. It is a symbolic regression to recover exact physical expression. For physical systems with known priors, we could use predefined function forms as candidates. Otherwise, we adopt a state-of-the-art model such as equation learner to recover the complex expression (Sahoo et al., 2018a). Specifically, we define split parameters to represent the hybrid representation of physics embedding and DipDNN: $f(\boldsymbol{x}) = \lambda_{\text{Phy}} f_1(\boldsymbol{x}) + \lambda_{\text{DipDNN}} f_2(\boldsymbol{x})$, where $\lambda_{\text{Phy}} + \lambda_{\text{DipDNN}} = 1, \lambda_{\text{Phy}}, \lambda_{\text{DipDNN}} > 0$. The hybrid models are trained simultaneously to minimize empirical errors and recover the underlying function. Since DipDNN is invertible, we obtain $\hat{\boldsymbol{x}} = f_1^{-1}(\boldsymbol{y})$ from the inverse counterpart and plug into the recovered physical function $f_2(\hat{\boldsymbol{x}})$. It can be used to verify the physical consistency of the forward approximation in DipDNN.

## 4.2 NUMERICAL REGULARIZATION

Even though the forward model is analytically invertible, numerical errors may be aggravated when computing the inverse solution. Here we show the inverse computation sensitivity on well-trained DipDNNs (MAPE $< 0.01\%$) of different depths using various synthetic datasets (details in A.4). We show in Fig. 4(a) the error propagation through layers compared with the ablation error of each invertible block via testing on synthetic datasets with different dimensions and nonlinearity.

We observe an exponential increase in the propagated error while the ablation error is nearly zero ($< 10^{-10}$). The numerical errors include round-off in Python implementation, forward approximation mismatches, data noises, etc. If the singular values of the forward mapping approach zero (without actually being zero, thus maintaining analytical invertibility), the singular values of the corresponding inverse mapping can become exceedingly large and amplify numerical errors, which is termed as an exploding inverse (Behrmann et al., 2021). Fig. 4(left) empirically shows that such errors will be aggravated and propagated as the problem size and network depth increase.

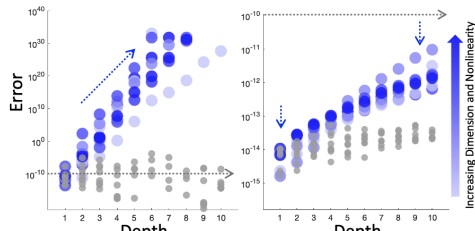

Figure 4: Compare the propagated errors (blue) through layers and the ablation errors (gray) without (left) and with (right) stability regularization.

We quantify the correlation between errors and inverse stability using bi-Lipschitz continuity with full details in Appendix A.4. Based on that, we enforce moderate regularization in the inverse mapping. For each layer, Leaky ReLU is a typical 1-Lipschitz activation function, and we adopt the $L_2$ norm of the inverse weights to smoothly clip large entries. While it is a moderate bound to regularize bi-Lipschitz continuity, the effect on the synthetic examples shows a much smaller error ($(< 10^{-10})$) propagated through layers in Fig. 4 (right).

## 5 RELATED WORK

### 5.1 DNN-BASED INVERSE LEARNING

Considering the approximation strategies, DNN-based inverse learning includes direct mapping recovery and two-way mapping recovery that unifies the pair of forward and inverse mappings. The inverse mapping is usually more complex than the forward (Kamyab et al., 2022). Thus, direct

mapping easily leads to overfitting and a mismatch in between. For example, unlike the physical priors of the forward system model, the inverse does not have a pre-defined physical form as a reference for interoperability (Raissi et al., 2019). Therefore, some studies combine forward and inverse learning together to match the bi-directional information flow (Arridge et al., 2019). There are various ways to realize such unified bi-directional learning: 1) minimizing the reconstruction errors to approximate a pair of forward and inverse mappings (Pakravan et al., 2021; Goh et al., 2019) and 2) enforcing invertibility in the forward model (Dinh et al., 2014; 2016; Ardizzone et al., 2018). For 1), the reconstruction error is unavoidable to ensure a matched inverse. As DNNs are not one-to-one mappings naturally, 2) includes invertible designs that either nest the DNNs in a triangular map or normalize the parameters for the Lipschitz constraint. The former can obtain an analytical inverse at the cost of stacking multiple layers with nested DNNs for full representation power, which aggravates error propagation (Dinh et al., 2014). The latter relaxes the restrictions on DNN architecture but relies on a fixed-point algorithm to approximate the inverse after forward training (Behrmann et al., 2019). The comparison of different invertible models shows there is a trade-off between the representation efficiency and inverse computation stability, which is also supported by theoretical analysis (Gottschling et al., 2020). In this paper, we make an attempt to minimize the adjustment on standard DNNs with respect to preserving the analytical inverse solution.

## 5.2 IDENTIFICATION-BASED STATE ESTIMATION

There are various inverse problems regarding the recovery of latent variables from physical measurements, e.g., vision-related tasks and extracting true states from observation of physical/engineering systems for monitoring and control (Gregor & LeCun, 2010; Engl et al., 1996; Benning & Burger, 2018). Traditional works solve such problems by iterative simulations, nearest search in a subspace, or optimization-based algorithms (Kucuk & Bingul, 2006; Tinney & Hart, 1967; 141, 1992; Pei et al., 2019). Typically, the identification-based state estimation differs from the traditional setting of state estimation, which has a completely accurate system model. Instead, it is a blind scenario where only measurements are available without knowing the full model (Liu et al., 2021; Haque et al., 2015; Liao et al., 2003). Therefore, previous work starts with model-free methods to approximate a direct mapping for state estimation (Chen et al., 2019). More works try to build in physical function in the forward mapping and conduct state estimation in the inverse simultaneously using a variational autoencoder (Goh et al., 2019; Dittmer et al., 2020; Hu et al., 2020; Pakravan et al., 2021). However, they do not enforce strict one-to-one correspondence for inverse consistency. Even though some generative models build bijectivity, the learning mechanism does not fit most of the discriminative learning tasks in physical/engineering systems, which have a more critical requirement on accurate point estimates for both in-distribution state restoration and extrapolation scenarios. Therefore, this paper aims to show that strict one-to-one mapping is possible with proper regularization.

## 5.3 REGULARIZATION FOR INVERSE LEARNING

The performance of inverse learning is challenged in both directions based on the accuracy-stability trade-off (Gottschling et al., 2020). Therefore, many regularization strategies are used to minimize extrapolation errors and ensure stable inverse reconstruction. Typically, for systems with prior knowledge, model-based regularizations include physics-informed deep learning via physics loss embedding (Stewart & Ermon, 2017; Kaptanoglu et al., 2021; Raissi et al., 2019; Bu & Karpatne, 2021), sparse symbolic regression yields law of parsimony (Occam's razor) (Brunton et al., 2016; Sahoo et al., 2018b), restrict relationships and dependencies between variables (Cotter et al., 2019; Udrescu & Tegmark, 2020; Fioretto et al., 2020; Zhao et al., 2019). While they solve specific problems effectively with strong priors, the predefined physics bias and portion may limit DNN's flexibility to choose the optimal representation. Besides, the regularization over the forward system identification may break the invertibility for inverse computation. Recent works attempt meta-algorithms to switch between a trusted physics agent and an untrusted black-box expert for robustness-accuracy balance in safety-critical control tasks (Li et al., 2022; Christianson et al., 2023). Such emerging research inspired us to design a twin structure to find the optimal integration of physics embedding and DipDNN approximation without hurting invertibility for general inverse problems.

## 6 EXPERIMENTS

In experiments, we test the capability of the proposed DipDNN on representative tasks, including synthetic examples, system identification-based state estimation, privacy-preserving learning, and image restoration. We aim to analyze the representation power and computation efficiency for forward mapping approximation and the inherent consistency of bi-directional mappings for inverse computation. Ablation studies are performed to understand better the model's restrictiveness and accuracy in practical implementation.

**Evaluation Metrics and Baseline Methods.** We use estimation errors to evaluate the forward approximation accuracy and inverse reconstruction/prediction of the bi-directional model via mean square error (MSE) and mean absolute percentage error (MAPE). For synthetic examples and physical systems, we further use the recovery rate (%) for the parameters or functional forms. The following methods are used in comparison: 1) *Autoencoder:* Autoencoders enforce invertibility in two DNNs with a reconstruction loss, which is used by many discriminative learning tasks for its flexible construction. We build the DNNs with the same architecture (depth, width, and activation) as DipDNN in each case. 2) *Addictive Coupling Layers:* The NICE model (Dinh et al., 2014) is designed for density estimation and trained with MLE using simple distribution sampling as inputs. In our case, we only build the invertible model and train it with MSE (Ardizzone et al., 2018). 3) *Invertible Residual Neural Network (i-ResNet):* While i-ResNet is similar to other generative models built on the probabilistic setup, we can use ResNet + Lipschitz constraint for discriminative learning. Its inverse is not analytically obtained from the forward model but needs an algorithm of fixed-point iteration (Behrmann et al., 2019). Training details are included in Appendix A.5.

**Synthetic Examples.** We use both synthetic datasets of symbolic functions and elementary physical functions from (Udrescu & Tegmark, 2020). The problem size is small (from 2 variables up to 9 variables), and the explicit functions are intuitive for demonstration. Sec. 4.2 presents a simple experiment for inverse stability. Fig. 5 shows that physics embedding improves the generalization with data scarcity and data variation.

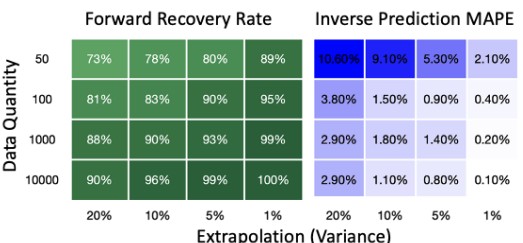

Figure 5: Correlate forward physical recovery rate (left) with inverse prediction error (right).

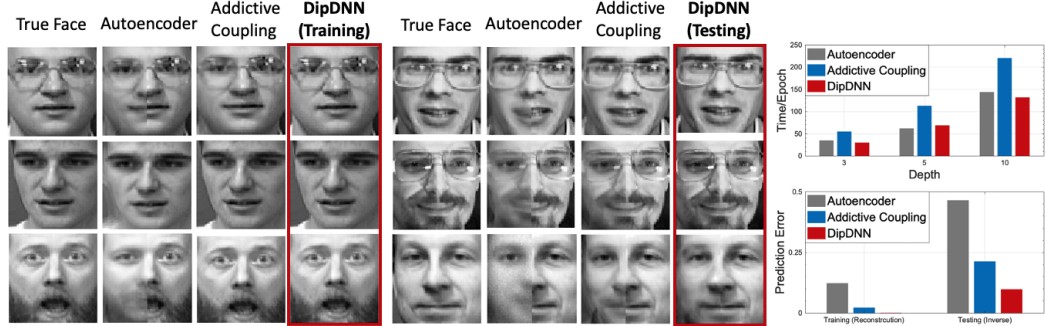

Figure 6: Visualization examples of the face image completion experiment.

**Image Construction and Face Completion.** The imaging-related tasks have much higher dimensions of input and output space, and the underlying mapping is difficult to interpret. MNIST (Deng, 2012; LeCun, 1998) has been used to test density estimation by NICE, transforming from a simple distribution (logistic prior) to complex images. Here, we adopt a discriminative setting to sample input data using logistic distribution. The NICE model, i-ResNet, and DipDNN are trained with MSE for 1000 epochs, and we mainly compare prediction errors, reconstruction errors, and computation time instead of log-likelihood in Fig. 7(b). With image dimension $28 \times 28 = 784$, we use the same architecture (MLP with 3-8 hidden layers and Leaky ReLU activation) for each model. For NICE, it is four coupling layers, with each containing one MLP. Moreover, we consider a representative bi-directional learning setup that $x$ and $y$ reveal similar features or nearly symmetric patterns, which

| Inverse | Autoencoder | NICE | i-ResNet | DipDNN | | Forward | MSE | Time (sec/epoch) |
|---------|-------------|------|----------|--------|---|---------|-----|------------------|
| PS(8-bus) | $0.06 \pm 0.03$ | $0.07 \pm 0.01$ | $0.007 \pm 0.00$ | $\mathbf{0.002 \pm 0.00}$ | | NICE | $0.234 \pm 0.089$ | $42.6 \pm 1.9$ |
| PS(123-bus) | $0.19 \pm 0.05$ | $0.20 \pm 0.02$ | $0.10 \pm 0.04$ | $\mathbf{0.01 \pm 0.00}$ | | i-ResNet | $0.0356 \pm 0.023$ | $13.4 \pm 2.7$ |
| SIR-inv | $0.37 \pm 0.17$ | $0.22 \pm 0.04$ | $0.34 \pm 0.07$ | $\mathbf{0.09 \pm 0.01}$ | | DipDNN | $0.0562 \pm 0.011$ | $16.9 \pm 7.8$ |
| Forward | Autoencoder | NICE | i-ResNet | DipDNN | | Inverse | MSE | Time (sec/epoch) |
| PS(8-bus) | $0.02 \pm 0.00$ | $0.04 \pm 0.01$ | $0.01 \pm 0.00$ | $\mathbf{0.02 \pm 0.00}$ | | NICE | $3.24 \times 10^{-6}$ | $4.1 \pm 1.5$ |
| PS(123-bus) | $0.11 \pm 0.01$ | $0.09 \pm 0.04$ | $0.12 \pm 0.03$ | $\mathbf{0.12 \pm 0.01}$ | | i-ResNet | $1.1 \times 10^{-5}$ | $24.2 \pm 3.9$ |
| SIR-fwd | $0.41 \pm 0.11$ | $0.21 \pm 0.07$ | $0.34 \pm 0.02$ | $\mathbf{0.36 \pm 0.09}$ | | DipDNN | $1.36 \times 10^{-10}$ | $1.6 \pm 0.2$ |

(a) Estimation errors (MSE$\times 10^{-3}$) of physical system state estimation.  (b) Results for MNIST.

Figure 7: Compare the forward prediction errors and inverse reconstruction with baseline methods.

need consistency in the forward-inverse pair. We use the classic face completion task to impaint the left face from the right (Pedregosa et al., 2011). The modified Olivetti face dataset (Roweis) is used. It consists of 10 pictures, each of 40 individuals, and each image is reformed in $64 \times 64$ grayscale. The images are separated into the left and right halves and reformed into vectors for learning. The visual results in Fig. 6 intuitively show the reconstruction of the left half given the right half. Compared with the blurring results of autoencoder, DipDNN reconstructs more details with analytical inverse, so as NICE model in training. The differences are more evident in unseen data. NICE model takes more time to build the same nonlinear couplings, and DipDNN tends to spend more time checking the satisfaction of invertibility at each iteration with increasing depth.

**System Identification-based State Estimation.** DipDNN can fit into various scenarios of state estimation. Fig. 8 and Fig. 7(a) show partial results of the following: (1) *Power System (PS) State Estimation:* It is an essential inverse problem to estimate voltage phasor states from standard measurements (e.g., power injections, branch power flows, and current magnitudes) (Hu et al., 2020; Sundaray & Weng, 2023).

(2) *False Data Injection Attacks:* User data are sensitive, and critical infrastructure can be vulnerable. On the defender side, operators need to understand attacking mechanisms to design countermeasures. By training a proxy autoencoder model, they incentivize the generation of tampered measurements that will produce indistinguishable measurements in data acquisition systems (Costilla-Enriquez & Weng, 2023). We collect data from real power systems and conduct simulations in different scenarios. Case details and more results are included in Appendix

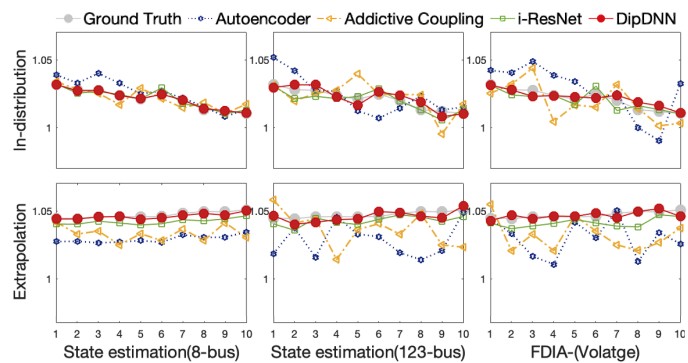

Figure 8: Validating state estimation results on system nodes.

A.5. (3) *Sonar performance analysis:* The signal-to-interference ratio (SIR) in the surveillance area at each pixel is a function of a number of parameters, including sonar depth, wind speed, bottom type, sound velocity, etc. (Jensen et al., 1999). With emulation data, DNNs are trained to map SIR pixel values from sonar and environmental parameters. The inverse problem is to quickly determine a set of input parameters that can yield a high SIR in the target area.

## 7 CONCLUSION

The proposed model can enforce strict one-to-one correspondence via relatively simple reconstructions of standard neural networks. We further show that it relaxes the computation burden of previous addictive coupling layers without hurting the universal approximation. Hence, it can better fit general inverse problems that target inverse consistency and discriminative point estimates of system states. Provided with certain prior knowledge and moderate stability regularization, the performance can be further boosted on both the forward approximation and inverse computation. This work explored only representatives of the possible applications of DipDNNs. The proposed designs, such as a twin structure with physics embedding, open the door for many additional domains that need a robust and consistent bi-directional information flow.

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

# A  APPENDIX

## A.1  DERIVATION FOR COMPOSITION OF ADDICTIVE COUPLING LAYERS

**Jacobian Derivation.**    For the $k^{th}$ unit of addictive coupling layers, the Jacobian is

$$J_k = \begin{bmatrix} \frac{\partial \boldsymbol{y}_{I_1}{}^{(k)}}{\partial \boldsymbol{x}_{I_1}{}^{(k)}} & \frac{\partial \boldsymbol{y}_{I_1}{}^{(k)}}{\partial \boldsymbol{x}_{I_2}{}^{(k)}} \\ \frac{\partial \boldsymbol{x}_{I_2}{}^{(k)}}{\partial \boldsymbol{x}_{I_1}{}^{(k)}} & \frac{\partial \boldsymbol{y}_{I_2}{}^{(k)}}{\partial \boldsymbol{x}_{I_2}{}^{(k)}} \end{bmatrix} = \begin{bmatrix} a^{(k)}I_1 & \boldsymbol{0} \\ \frac{\partial \boldsymbol{t}^{(k)}(\boldsymbol{x}_{I_1}{}^{(k)})}{\partial \boldsymbol{x}_{I_1}{}^{(k)}} & b^{(k)}I_2 \end{bmatrix}. \tag{4}$$

For every other layer, the columns exchange due to the in-turn composition in Fig. 1(b). Using the chain rule, the Jacobian of the composited function is $\Pi_k J_k$. Only when $k \geq 3$, the 0's can be eliminated from the Jacobian matrix, and all the elements can be non-constant, thus indicating a full transformation of all dimensions.

$$\Pi_3 J_3 = \begin{bmatrix} a_3 I_1 & \boldsymbol{0} \\ \frac{\partial \boldsymbol{t}_3}{\partial \boldsymbol{z}_{I_1}^2} & b_3 I_2 \end{bmatrix} \cdot \begin{bmatrix} a_2 I_1 & \frac{\partial \boldsymbol{t}_2}{\partial \boldsymbol{z}_{I_2}^1} \\ \boldsymbol{0} & b_2 I_2 \end{bmatrix} \cdot \begin{bmatrix} a_1 I_1 & \boldsymbol{0} \\ \frac{\partial \boldsymbol{t}_1}{\partial \boldsymbol{x}_{I_1}} & b_1 I_2 \end{bmatrix} \tag{5}$$

$$= \begin{bmatrix} a_3 a_2 a_1 I_1 + a_3 \frac{\partial \boldsymbol{t}_2}{\partial \boldsymbol{z}_{I_2}^1} \frac{\partial \boldsymbol{t}_1}{\partial \boldsymbol{x}_{I_1}} & a_3 b_1 \frac{\partial \boldsymbol{t}_2}{\partial \boldsymbol{z}_{I_2}^1} \\ a_2 a_1 \frac{\partial \boldsymbol{t}_3}{\partial \boldsymbol{z}_{I_1}^2} + b_3 b_2 \frac{\partial \boldsymbol{t}_1}{\partial \boldsymbol{x}_{I_1}} + \frac{\partial \boldsymbol{t}_3}{\partial \boldsymbol{z}_{I_1}^2} \frac{\partial \boldsymbol{t}_2}{\partial \boldsymbol{z}_{I_2}^1} \frac{\partial \boldsymbol{t}_1}{\partial \boldsymbol{x}_{I_1}} & b_3 b_2 b_1 I_2 + b_1 \frac{\partial \boldsymbol{t}_3}{\partial \boldsymbol{z}_{I_1}^2} \frac{\partial \boldsymbol{t}_2}{\partial \boldsymbol{z}_{I_2}^1} \end{bmatrix}. \tag{6}$$

Fig. 1(b) shows intuitively that the second layer completes the nonlinear coupling between $\boldsymbol{x}_{I_1}$ and all $\boldsymbol{y}$. However, as we highlight the paths in gray, $\boldsymbol{x}_{I_2}$ has the first linear path towards $\boldsymbol{y}_{I_2}$ and still needs the third layer to nonlinearly map to $\boldsymbol{y}_{I_2}$. Assuming we use the same architecture (i.e., $K$-layer ReLU-activated DNNs) for the nonlinear function in each layer, Fig. 1(b) needs three such DNNs to complete the coupling among all dimensions. The separate DNNs shall weaken the approximation capability so that four or more layers are usually concatenated in implementation.

**Representation Efficiency and Computation Complexity.**    In Sec. 2.2, we show the typical reconstruction to enforce strict invertibility. As we aim to minimize the adjustment on DNNs subject to invertibility, we compare its representation power with regular DNNs. To compare with the dense fully connected layers, Fig. 9(c) needs to be compressed further. To achieve this, we will have to superpose the two layers as shown in Fig. 9(d). It can be seen as an equivalent architecture in representation, e.g., the same nonlinear correlations between each input/output group. The unfolded

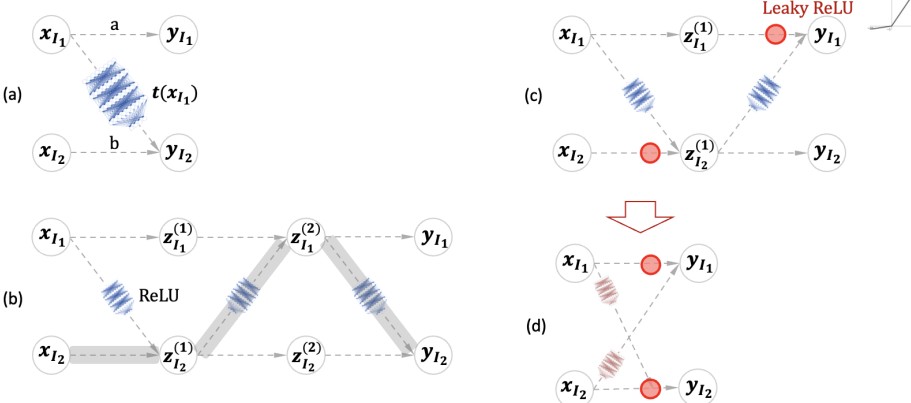

Figure 9: (a) - (b): Composition of addictive invertible transformation for full coupling of input/output dimensions. (c): A reduction of (b) that retains full dimension coupling. (d) A further reduction of (c) for equivalent representation.

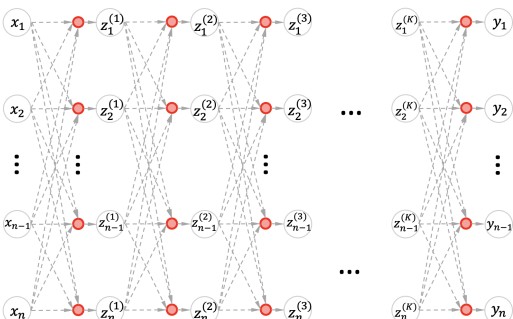

Figure 10: The structure of regular DNN with Leaky ReLU activation function.

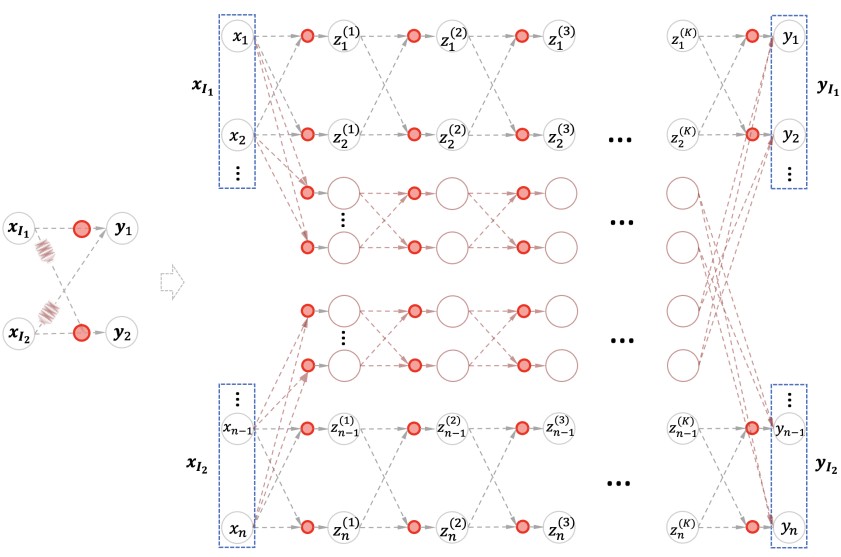

Figure 11: Unfold Fig. 9(d) as a sparse DNN.

model in Fig. 11 can be seen as a sparse connection of regular DNN in Fig. 10. Many interconnections are eliminated due to the separation of input/output groups, i.e., the full connectivity is isolated within

each group coupling. Thus, the invertibility is enforced at the cost of $\sim 1.5\times$ computation complexity and sparse interconnections among different groups of variables.

## A.2 PROPOSED DIPDNN

**Basic Invertible Unit.** Motivated by the triangular map, the neural representation of one invertible unit (top corner of Fig. 2) is

$$
\begin{aligned}
\boldsymbol{z}^{(1)} &= \text{LeakyReLU}(W_{tril}\boldsymbol{x} + \boldsymbol{b}_1), \\
\boldsymbol{z}^{(2)} &= \text{LeakyReLU}(W_{triu}\boldsymbol{z}^{(1)} + \boldsymbol{b}_2),
\end{aligned}
\tag{7}
$$

where $W_{tril} = \begin{bmatrix} w_{11}^{(1)} & 0 \\ w_{21}^{(1)} & w_{22}^{(1)} \end{bmatrix}, W_{triu} = \begin{bmatrix} w_{11}^{(2)} & w_{12}^{(2)} \\ 0 & w_{22}^{(2)} \end{bmatrix}$. Subsequently, the corresponding inverse function is

$$
\begin{aligned}
\boldsymbol{z}^{(1)} &= W_{triu}^{-1}(\text{LeakyReLU}^{-1}(\boldsymbol{z}^{(2)}) - \boldsymbol{b}^{(2)}), \\
\boldsymbol{x} &= W_{tril}^{-1}(\text{LeakyReLU}^{-1}(\boldsymbol{z}^{(1)}) - \boldsymbol{b}^{(1)}).
\end{aligned}
\tag{8}
$$

The unit equation 7 is strictly invertible if $w_{11}^{(1)}, w_{22}^{(1)}, w_{11}^{(2)}, w_{22}^{(2)} \neq 0$, and the inverse is computed in equation 8.

**Supplementary Theorems.** Details of Theorem 2 are as follows.

**Theorem 3.** *Let $\mathcal{K} \subset \mathbb{R}^{dx}$ be a compact set; then, for the continuous function class $C(\mathcal{K}, \mathbb{R}^{dy})$, the minimum width $w_{min}$ of Leaky-ReLU neural networks having $C - UAP$ is exactly $w_{min} = \max(dx + 1, dy) + 1 = dx + 1 + dy$. Thus, $NN(\sigma)$ is dense in $C(K, \mathbb{R}^{dy})$ if and only if $N \geq w_{min}$.*

$$
1_{dy=dx+1} = \begin{cases} 1, & \text{if } dy = dx + 1, \\ 0, & \text{if } dy \neq dx + 1. \end{cases}
$$

**Lemma 1.** *For any continuous function $f^* \in C(\mathcal{K}, \mathbb{R}^d)$ on a compact domain $\mathcal{K} \subset \mathbb{R}^d$, and $\epsilon > 0$, there exists a Leaky-ReLU network $f_L(x)$ with depth $L$ and width $d + 1$ such that*

$$
\|f_L(x) - f^*(x)\| \leq \epsilon
$$

*for all $x$ in $\mathcal{K}$.*

As a follow-up of Theorem 3, Lemma 1 specifies the condition for the case where the input and output dimensions are equal, $d_x = d_y = d$. It provides the result that the Leaky ReLU-activated neural network with width $d + 1$ has enough expressive power to approximate the continuous function $f^*$.

The theorem on the identity mapping approximation is used to prove the decomposition of weight matrices.

**Theorem 4.** *Let $I : \mathbb{R}^n \to \mathbb{R}^n$ be the identity map. Then for any compact $\Omega \subseteq \mathbb{R}^n$ and any $\epsilon > 0$, there exists a $\delta > 0$ such that whenever $0 < |h| < \delta$, the function $\rho_h : \mathbb{R}^n \to \mathbb{R}^n$, satisfies*

$$
\rho_h(x) := \frac{1}{h\sigma'(a)}[\sigma(hx + a\mathbf{1}_n) - \sigma(a)\mathbf{1}_n],
$$

*and*

$$
\sup_{x \in \Omega} \|\rho_h(x) - I(x)\| \leq \epsilon.
$$

## A.3 MODEL USED FOR PHYSICS EMBEDDING

Fig. 12 is a toy example to intuitively show the importance of physical guidance. See Fig. 13 for the physics embedding model. For systems with prior functional forms, the passthrough is activated to avoid complex symbolic regression. Otherwise, we use equation learner (Sahoo et al., 2018b) to recover explicit function.

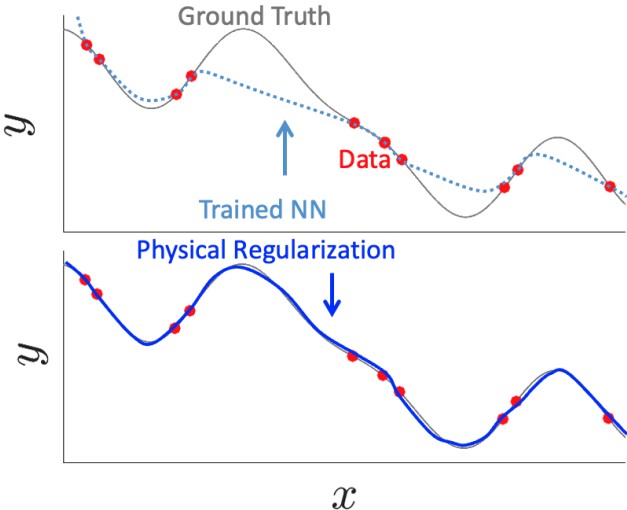

Figure 12: A toy example to motivate physics embedding.

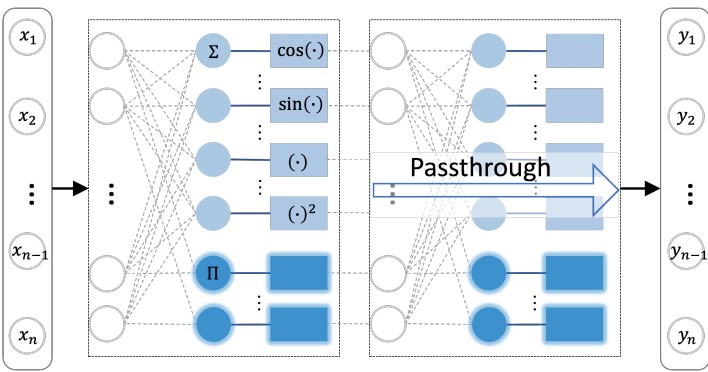

Figure 13: The model to recover physical symbolic form and provide physical regularization over the DipDNN.

### A.4 INVERSE STABILITY

**Toy Examples.** Here, we test the performance of the inverse mapping based on a well-trained invertible neural network model (forward error $\approx 0$). For a comprehensive understanding, we test on cases with different setups: 1) $x, y \in \mathbb{R}^m, m \in \{2, 4, 6, 8\}$, 2) $f$ is built based on the mix of different basis $\{(\cdot), (\cdot)^2, \cos(\cdot), \sin(\cdot), \exp(\cdot)\}$, 3) different depth of network $K = \{2, 4, 6, 8, 10\}$. We select representative examples and show in Fig. 4 the error propagation through layers compared with the ablation error of each invertible block.

**Lipschitz and Bi-Lipschitz Continuity related to Inverse Stability** The numerical invertibility is related to Lipschitz continuous property for mapping contraction. As for the Lipschitz continuity of both the forward and inverse mapping, we recall the definition of bi-Lipschitz continuity of invertible function.

**Definition 1.** *(Lipschitz and bi-Lipschitz continuity of invertible function) A function $f : \mathbb{R}^n \to \mathbb{R}^n$ is called Lipschitz continuous if there exists a constant $L := Lip(f)$ such that:*

$$\|f(x_1) - f(x_2)\| \leq L\|x_1 - x_2\|, \forall x_1, x_2 \in \mathbb{R}^n. \tag{9}$$

*If an inverse $f^{-1} : \mathbb{R}^n \to \mathbb{R}^n$ and a constant $L^* := Lip(f^{-1})$ exists, then, for all $y_1, y_2 \in \mathbb{R}^n$,*

$$\|f^{-1}(y_1) - f^{-1}(y_2)\| \leq L^*\|x_1 - x_2\|, \forall x_1, x_2 \in \mathbb{R}^n. \tag{10}$$

*Thus, the function $f$ is called bi-Lipschitz continuous.*

From the mathematical definition, the Lipschitz constant reflects the amplification of errors in the worst scenario during the reconstruction. For example, i-ResNet adopts a strict bound $\text{Lip}(f) < 1$ as a sufficient condition of invertibility and enforces stable inverse computation via fixed-point iterations. But, recent theoretical results of no free lunches in (Gottschling et al., 2020) show an accuracy-stability trade-off in inverse problems: high stability may be reached at the expense of significantly poor approximation performance. In our case, DipDNN aims for the analytical inverse without requiring a rigid Lipschitz bound. Meanwhile, we have proposed physics embedding to avoid overperformance (only minimizing errors) on the training set. In order to further balance performance and stability, we quantify the correlation between errors and the Lipschitz bound.

**Correlation between Numerical Errors and Lipschitz Bounded Reconstruction Errors**   Let $\sigma$ denote the numerical errors (round-off, forward approximation errors, noises, etc.), then a forward mapping $z = h(x)$ becomes $h_\sigma(x) = z + \sigma = z_\sigma$. Subsequently, the inverse for input reconstruction becomes $x_{\sigma_1} = h^{-1}(z_\sigma)$. According to Definition 1, with Lipschitz constant $\text{Lip}(h^{-1})$, we derive the bound of reconstruction error

$$\begin{aligned}
\|x - x_{\sigma_1}\|_2 &\leq \text{Lip}(h^{-1})\|z - z_\sigma\|_2 \\
&= \text{Lip}(h^{-1})\|\sigma\|_2.
\end{aligned} \tag{11}$$

This is for errors propagated and aggravated from the forward mapping. As for the inverse mapping, we use $\sigma_2$ to denote errors for round-off and extrapolation (distribution shift), and get $h_\sigma^{-1}(z_\sigma) = x_{\sigma_1} + \sigma_2 = x_{\sigma_2}$. Similarly,

$$\begin{aligned}
\|x - (x_{\sigma_1} + \sigma_2))\|_2 &\leq \|x - x_{\sigma_1}\|_2 + \|\sigma_2\|_2 \\
&\leq \text{Lip}(h^{-1})\|z - z_\sigma\|_2 + \|\sigma_2\|_2 \\
&= \text{Lip}(h^{-1})\|\sigma\|_2 + \|\sigma_2\|_2,
\end{aligned} \tag{12}$$

which shows how much the Lipschitz constant amplifies the errors. As we observe empirically, $\sigma, \sigma_2$ will be increased with the problem becoming more complex. As long as we moderately regularize the $\text{Lip}(h^{-1})$, the reconstruction error can be bounded.

**Proof of Bi-Lipschitz Continuity of Inverse Mapping**   If we can ensure the bi-Lipschitz continuity of an invertible function, the corresponding inverse mapping is also bi-Lipschitz continuous. The proof is simple in that, with equation 9, $f$ has one-to-one correspondence $f(x_1) = f(x_2) \Leftrightarrow x_1 = x_2$. With equation 10, for all $y_1, y_2 \in f(\mathcal{X})$, there is a unique $x_1, x_2 \in \mathcal{X}$ such that $f^{-1}(y_1) = x_1$ and $f^{-1}(y_2) = x_2$. By substituting them into equation 9 and equation 10, we obtain $\|y_1 - y_2\| \leq L\|f^{-1}(y_1) - f^{-1}(y_2)\|$, meaning bi-Lipschitz of $f^{-1}$. Considering equation 11 and equation 12, with bounded Lipschitz constant throughout the network, we can guarantee stable performance in two-way mapping numerically.

Therefore, we enforce moderate Lipschitz continuity in the inverse mapping. The exact computation of DNN's Lipschitz bound is NP-hard, but we can decompose it into layers with $\text{Lip}(h^{(k-1)} \circ h^{(k)}) = \text{Lip}(h^{k-1}) \cdot \text{Lip}(h^k)$. For each layer, Leaky ReLU is a typical 1-Lipschitz activation function, so we need to constrain the parameters of the inverse form to avoid arbitrarily large values. Both $L_1$ and $L_2$ norms can mitigate the weights explosion, but the $L_1$ norm simultaneously encourages the sparse connection of the network. In our case, the inverses of triangular weight matrices retain their triangular form and inherently exhibit sparsity. Applying $L_1$ norm regularization could potentially hinder training efficiency and violate the conditions for invertibility. Therefore, we adopt the $L_2$ norm of the inverse weights to smoothly clip large entries. While it is a moderate bound to regularize bi-Lipschitz continuity, the effect on the synthetic examples shows a much smaller error $((< 10^{-10}))$ propagated through layers in Fig. 4 (right).

## A.5   EXPERIMENTS

**Training Details.**   The Pytorch platform is used to build and train models by Adam optimizer for $> 200$ epochs for each experiment, and early stopping is used. Hyperparameters such as the number of layers (blocks), negative slope coefficient of Leaky ReLU ($\{0.1, 0.2, 0.05\}$), learning rate ($\{0.001, 0.0002, 0.00005\}$), weighting factor for regularization ($\{0, 0.1, 0.5, 1\}$) and dropout rate, are adjusted by grid search for each dataset on a validation set. All the experiments are implemented

on a computer equipped with Inter(R) Core(TM) i7-9700k CPU and Nvidia GeForce RTX 3090 GPU. Each experiment is run at least 3-5 times to record the average results and statistics for a fair comparison.

As for DipDNN, we enforce non-zero entries on the diagonal of lower/upper triangular matrices to ensure no violation of invertibility. Previous work on restrictive DNN of input convexity uses a post-check on each training iteration, e.g., replacing negative weights with the opposite to enforce positive weights (Amos et al., 2017; Chen et al., 2018). Empirically, we find this way takes effect but slows down the training and may lead to non-convergence in large systems, with the parameters getting trapped at zero sometimes. Instead, we add a mask (lower or upper triangular) over the weight matrix, which alleviates the parameter fluctuations in experiments.

**Details of Data Preparation for Physical Systems.**  Compared to the elementary physical systems, the power system has a much larger problem size and more complex variable couplings due to the grid connection. Moreover, to fulfill the need for downstream control and operations, the learning model needs to have robust performance in extrapolation scenarios, such as increasing load and renewable injection conditions. For the forward, the governing physical law is the grid power flow equations expressed by system-wide power injections based on voltage phasors and system topology/parameters (Yuan & Weng, 2021). Traditional methods assume an available model and linearize the model to solve for the states. However, system parameters are unknown or incorrect in many secondary power distribution systems (Cavraro & Kekatos, 2018; Cavraro & Kekatos, 2019; Yuan et al., 2016; Moffat et al., 2020).

IEEE provides standard power system models (IEEE 8- and 123-bus test feeders), including the grid topology, parameters, generation models, etc., for simulations. The model files and the simulation platform, MATPOWER (MATPOWER, 2020), are based on MATLAB. For simulations, the load files are needed as the inputs to the systems. Thus, we introduce power consumption data from utilities such as PJM Interconnection LLC (PJM Interconnection LLC, 2018). Such load files contain hourly power consumption in 2017 for the PJM RTO regions. For the Utility feeder, the collected data also includes distributed photovoltaics generation profiles. With the above data, MATPOWER produces the system states of voltages and nodal power injections. Furthermore, the loading conditions and renewable penetration keep changing in the current power systems. We validate the extrapolation capability of the proposed invertible model using out-of-distribution data ($3\times$ PV generation and loads).

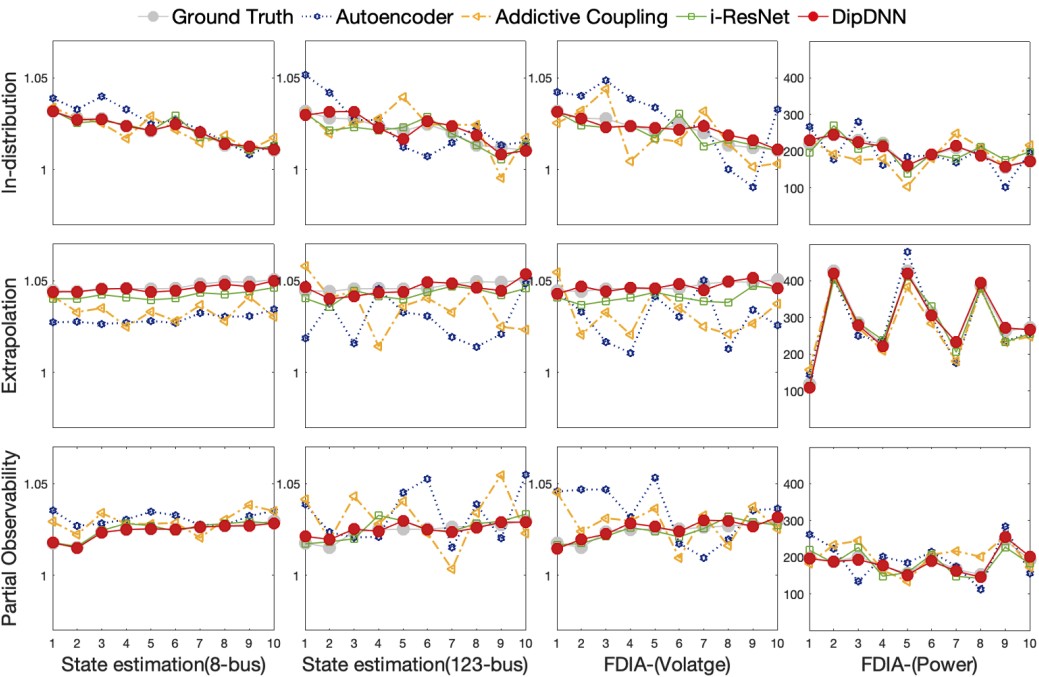

Figure 14: Validating state estimation results in physical system node.

