# OpenReview forum: "DipDNN: Decomposed Invertible Pathway Deep Neural Networks"
_ICLR.cc/2024/Conference — Submitted to ICLR 2024_

### Official Review · Reviewer_zeZn · 2023-10-29

**Soundness:** 2 fair
**Presentation:** 1 poor
**Contribution:** 1 poor
**Rating:** 3
**Confidence:** 4

**Summary:**

The authors present a novel model DipDNN for the intent of solving bi-directional inference problems. The model enforces bijectivity by using coupling layers, inspired by the work of  Dinh et al., 2014, to construct an analytical inverse, which is easy to compute and does not restrict the model's capacity shown in theorem 1. In addition, the proposed method reduces computation time by utilizing upper and lower triangular weight matrices in the model, which is demonstrated in Figure 2. Lastly, the proposed method is empirically validated for a variety of inverse problems, including image construction competition and system identification-based state estimation.

**Strengths:**

1). Diversity in types of experiments, the authors evaluated the proposed wide range of applications.

2). The results of the proposed method are empirically promising in the system identification-based state estimation experiments.

**Weaknesses:**

1). The paper lacks some important details that make the paper challenging to evaluate.
*  The training objective of the proposed method is not articulated clearly mathematically. The paper should clearly state the objective
    function and what parameters are being optimized.
* Objective function defined in section 2.1 the set g is an element of is not defined.
* Figures 1 and 2 lack a legend/an explanation of the operations. What are the blue lines and red dots?
* Similar to training the objective function for inference is not stated. How are the inference problems being solved? What is the mathematical objective?

2). Authors should provide an explanation of why they chose an additive coupling layer over coupling methods, i.e. Affine or spline coupling layers. Normalizing flow models have made significant progress in the area of coupling layers, so the authors should justify why they're choosing additive over another type.

3). Novelty of theoretical contribution
* The results of Theorem 1 are heavily dependent on the contribution of Duan et al., 2023.
* Coupling layers have been proven to be universal approximations [Coupling-based Invertible Neural Networks Are
Universal Diffeomorphism Approximators](https://proceedings.neurips.cc/paper/2020/file/2290a7385ed77cc5592dc2153229f082-Paper.pdf) Nuerips 2020


4). Image Construction and Face Completion experiments
* Figure 6 Based on the eye test it is challenging to see the improvements in the DipDNN over the Additive Coupling
* Based on Figure 7 Results B i-Resnet outperforms the proposed method, but the authors do not make a comment about this result in the paper.

**Questions:**

Questions are addressed in the weakness section.

Main questions:
* Could the authors state the objective functions for both training and inference of the proposed method?
* Could authors please state the theoretical contributions and the insight they provide to the paper?
* Please provide justification or explanation on i-Resnet outperforming the proposed method in Figure 7 b.

---

> ### Author Response · Authors · 2023-11-23
> **Response to reviewer zeZn:**
>
> We would like to thank you for your time and effort in reviewing our paper. We are pleased you recognize the proposed method's applicability to diverse inverse problems. The suggestions you provided for clarifying the paper's target and contribution are very useful, and we will use them for revision. Below, please find our responses to the questions.
>
> **Specify objectives of the inverse problem:**
>
> *Define the inverse problem:*
>
> For system identification, the target is to learn a mapping $f: \mathcal{X} \rightarrow  \mathcal{Y}$ from data for estimating $y$ given $x$. The inverse problem is an opposite direction of system identification to infer states $x$ given $y$. Therefore, our target is first to learn a forward mapping and then derive the inverse counterpart. Mathematically, the objective function includes 1) a regular supervised learning error term of the forward learning and 2) a reconstruction error term of using the inverse mapping to recover back to $x$.
>
> *Clarify the mathematical objectives:*
>
> Thanks for pointing out the typo in the math expression. We checked it and found that the previous objective and description may cause confusion. We rewrite it here.
>
>
> As computing the inverse solution is the final target, we define the objective using the inverse mapping $g: \mathcal{Y} \rightarrow  \mathcal{X}$.
>
> $$ {g}^* =   \operatorname*{argmin}_{g \in \mathcal{G} } \sum_{i=1}^N \ell_1\left({g_{\theta}}^{-1}\left(\bm{x}_i\right),\bm{y}_i\right) + \sum_{i=1}^N \ell_2\left(\bm{x}_i, g_{\theta}\left({g_{\theta}}^{-1}\left(\bm{x}_i\right)\right)\right)$$
>
> The first term is the supervised learning loss to minimize the mismatch in forward model recovery, where $\ell_1(\cdot)$ denotes the mean square error we used in this paper. The second term is the reconstruction error to make the inverse consistent with the forward.
> $\theta$ are neural network parameters of the forward model, which is optimized during training. In our proposed method, we design analytical invertibility in the forward NN model such that the inverse counterpart shares the parameters $\theta$.
>
> Therefore, once the forward model is well-trained, we directly use the invertible NN to derive the inverse model and use it for inference of $x$. No more training is needed. This is also why an analytical inverse form is essential; otherwise, the inverse computation is challenging.
>
> *Clarify components in Figures 1 and 2:*
>
> We use Figure 1 to illustrate the architecture limitation and representation power of existing invertible networks. They motivate us to design the proposed DipDNN in Figure 2. Figure 1(a) is a basic layer of the classic NICE model [1]. The blue component denotes the nonlinear NN correlation, which is only between $x_{I_1}$ and $y_{I_2}$. ${x}_{I_1}$ and ${y}_{I_1}$, ${x}_{I_2}$ and ${y}_{I_2}$ are linearly correlated and $x_{I_2}$ and $y_{I_1}$ are not correlated. Therefore, at least three such layers are required to enable full nonlinear coupling of all inputs with all outputs in Figure 1(b). Regular NN takes one layer for full coupling, so we try to compress Figure 1(b) with an equivalent representation and compare the network connection with a regular NN. Red dots denote Leaky ReLU activation, providing nonlinearity on the linear paths in the reduced architecture. Leaky ReLU activation is a strictly monotone function customized from ReLU, which preserves invertibility in nonlinear mapping.
>
> Figure 2 presents the proposed decomposed-invertible-pathway DNN (DipDNN) model. The blue lines indicate connections in NN, and the red dots still denote Leaky ReLU. Here, we first maintain the dense NN architecture and then enforce one-to-one correspondence by decomposing the nested layer of NICE into a basic invertible unit, as shown in the top right corner of Figure 2. Subsequently, we extend the invertible unit to wider and deeper NN.

---

> > ### Author Response · Authors · 2023-11-23
> > **Response to reviewer zeZn (continued #1):**
> >
> > **Clarify the contributions with respect to existing methods:**
> >
> > The problem setup is different from that of most normalizing flow models, which are usually used for generative learning. For example, the image density estimation tasks aim to map from a Gaussian distribution to a complex image distribution. With the probabilistic setup to map from $p(x)$ to $p(y)$, the change of variables theorem can be used to build invertibility and maximum likelihood is used to train the model.
> >
> > However, the inverse problem is wider than generative tasks. Many applications target accurate point estimates, especially in physical and engineering systems. Directly applying SOTA coupling layers leads to poor performance. Therefore, rather than the invertibility defined between distribution densities, we need a deterministic invertible mapping function to estimate the inverse solution.
> >
> > However, a neural network is not invertible in nature because the multi-layers of interconnections break a strict one-to-one correspondence. We choose the additive coupling layer [1] because its NN architecture is not only strictly bijective but also has an easy-to-derive analytical inverse form. As the reviewer mentioned, the following works mainly focus on improving the coupling layers for generative tasks, and few improve the bijective mapping function. So, they are inapplicable in our inverse problem setup. For similar applications on discriminative learning tasks, the latest variation of affine coupling is [2]; we also refer to their training details in experiments.
> >
> > Moreover, the affine coupling layers motivate our DipDNN for discriminative inverse problems.
> > We initially apply this model to the inverse problem in physical systems and found poor performance in that 1) the maximize likelihood loss function does not apply as our task is discriminative learning and the data is not generated from a Gaussian distribution, 2) the random splitting of image pixels is not fair for variables with physical meaning as different splitting groups lead to different results, 3) each layer ensures the invertibility via linear paths and bypasses nonlinear mappings. So, in this work, we evaluate how the proposed designs in DipDNN resolve the drawbacks.

---

> > > ### Author Response · Authors · 2023-11-23
> > > **Response to reviewer zeZn (continued #2):**
> > >
> > > **Justify the novelty of theoretical contribution:**
> > >
> > > First, we agree with the reviewer that Theorem 1 depends on the latest results to guarantee the universal approximation capability of narrow deep neural network. However, the theoretical contribution of this paper is to show that DipDNN enforces invertibility in dense DNNs without hurting the representation power of the DNN. In the following, we illustrate theoretical novelty based on the comparison with existing invertible NNs.
> > >
> > > We have illustrated the drawbacks in affine coupling layers above. Points 2) and 3) are related to the architecture limitation and representation power of existing invertible networks. To illustrate in detail with an intuitive example, we would like to point the reviewer to Appendix A.1, where Page 14 includes three figures to compare the network architectures. As a result, the nested design in the NICE model ensures invertibility at the cost of $\sim 1.5\times$ computation complexity (i.e., at least three layers for full coupling) and representation power (i.e., an equivalent sparse representation of regular NN). This is why we are motivated to design an invertible model that relaxes the fixed splitting of variables and maintains regular NN’s representation power as much as possible.
> > >
> > > In the proposed decomposed-invertible-pathway DNN (DipDNN) model, we first maintain the regular dense NN architecture and then enforce one-to-one correspondence by decomposing the nested layer of NICE into a basic invertible unit, as shown in the top right corner of Figure 2. Therefore, the designs of triangular weights and Leaky ReLU activation maintain invertibility in the nonlinear correlation of the NN. With invertibility, the remaining question is: Can we maintain representation power? The challenges are two-fold: some weights are not activated and the NN is narrow.
> > >
> > > For the former challenge, the triangular weights can be seen as an equivalent adjustment using LU decomposition of the regular weights if the activation function is linear. In the proof of Theorem 1, we further prove that using a nonlinear activation function retains the universal approximation capability for NNs with alternative lower and upper triangular matrices. The proof is based on a recent theoretical analysis of the decomposition of NNs [4].
> > >
> > > For the latter challenge, the proposed DipDNN is narrow, which is necessary to maintain one-to-one correspondence for invertibility. However, we can make it arbitrarily deep. Many efforts have been made recently to improve the universal approximation capability of narrow deep NNs. As the reviewer mentioned, the latest results on the minimum width of narrow deep NNs and the generally applicable theory in [3] serve as the basis of DipDNN’s theoretical guarantee. Before these results are established, we need to construct a wider NN with zero padding for universal approximation.
> > >
> > > *Regarding the universal approximation of coupling layers:*
> > >
> > > Thanks for suggesting the reference paper [6]. We initially referred to this paper. However, the paper focuses on the perspective of normalizing flows. The main results are derived for universal diffeomorphism approximation. For example, distributional universality is proven to guarantee probability distribution models constructed using invertible neural networks. Thus, the problem setup differs from ours, which we explained above in **Clarify the contributions with respect to existing methods:**. Moreover, the affine coupling layers take more computation effort to reach a similar approximation power.
> > >
> > > While the paper's conclusion is inapplicable, we are motivated by this paper to derive our theoretical analysis.
> > >
> > > [1] Dinh et al. “NICE: Non-linear Independent Components Estimation.” ICLR 2015.
> > >
> > > [2] Ardizzone et al. “Analyzing Inverse Problems with Invertible Neural Networks.” ICLR 2019.
> > >
> > > [3] Duan et al. “Minimum width of leaky-relu neural networks for uniform universal approximation.” ICML 2023.
> > >
> > > [4] Liu et al. “Lu decomposition and toeplitz decomposition of a neural network.” arXiv 2022.
> > >
> > > [5] Zhang et al. “Approximation capabilities of neural odes and invertible residual networks.” ICML 2020.
> > >
> > > [6] Teshima et al. “Coupling-based Invertible Neural Networks Are Universal Diffeomorphism Approximators.” Neurips 2020.

---

> > > > ### Author Response · Authors · 2023-11-23
> > > > **Response to reviewer zeZn (continued #3):**
> > > >
> > > > **Explain the results in experiments:**
> > > >
> > > > *Regarding face completion task:*
> > > >
> > > > Thanks for pointing out the presentation of the experiment results. For Figure 6, we agree with the reviewer that the visual difference is not obvious. The face completion dataset is relatively simple so that both models can have good results. We keep these results because estimating from the right-half face to the left-half face is more intuitive than regular image density estimation that maps from noises to images. The two-half faces share similar features, which is reasonable in our inverse problem setup focusing on point estimation. On the right-hand side of Figure 6, the computed prediction errors and running time can better show the improvement using DipDNN.
> > > >
> > > >
> > > > *Regarding comparison with i-Resnet:*
> > > >
> > > > Thanks for the chance to let us explain these results in detail. Considering the target of the inverse problem, several aspects need to be evaluated. The results in Figure 7(b) include: 1) prediction error of the forward learning, 2) reconstruction error of using the inverse mapping for inference, 3) running time for the forward training and 4) running time for inverse solution computation. For 1), i-Resnet and DipDNN outperform NICE, and i-Resnet is slightly better. This is reasonable as the i-Resnet is built upon Resnet, for which the residual block design has superior performance in forward learning. However, if we compare the inverse computation, DipDNN performs better. This is because i-Resnet does not have an analytical inverse form and needs fixed-point iterations to compute the inverse after training the forward model. On the contrary, DipDNN has an analytical inverse for direct computation. Thus, the inverse prediction error is lower and the inverse computation time is less for DipDNN.
> > > >
> > > >
> > > > *Extra test on complex dataset:*
> > > >
> > > > We add some experiments to test the model on the CIFAR-10 dataset and compare it with i-Resnet. We test a CIFAR-10 classification task and note that the forward-inverse mapping is built between inputs to the last feature extraction layer, where a linear classification layer is used on top of the invertible blocks. Due to the time limit, we reduce the hidden layers and control all the models to have the same number of layers (modes using MLP and having the dimension be 3072) for a fair comparison.
> > > >
> > > > | Model                        | i-Resnet(Conv)                                  | i-Resnet(MLP)          | NICE(MLP)          | DipDNN            |
> > > > |------------------------------|-------------------------------------------------|------------------------|--------------------|-------------------|
> > > > | Classification Acc. (Forward)| 74.85% (93.07% in [3] using full model)        | 69.73%                | 57.31%             | 65.23%            |
> > > > | Running time (Forward Training)| 26.27 Sec/Epoch                                | 14.13 Sec/Epoch       | 47.93 Sec/Epoch    | 12.46 Sec/Epoch   |
> > > > | Reconstruction Err. (Inverse)| 6.95e-5                                         | 8.32e-4               | 4.98e-3             | 3.27e-10          |
> > > > | Running time (Inverse Testing)| 45.81 Sec                                      | 37.64 Sec             | 5.23 Sec            | 2.46 Sec          |
> > > >
> > > > We can draw similar conclusions to the previous insights on MNIST results.
> > > >
> > > > Thanks again for the comments to let us review and clarify several contents of the paper.
> > > > Please let us know if our response addresses your comments. We would be happy to discuss any follow-up questions you may have!

---

### Official Review · Reviewer_uKYL · 2023-11-02

**Soundness:** 3 good
**Presentation:** 3 good
**Contribution:** 3 good
**Rating:** 6
**Confidence:** 3

**Summary:**

Targeted on the bi-directional inference demands, such as state estimation, signal recovery, privacy preservation, and reasoning, the paper presents a novel Deep Neural Network design, decomposed-invertible-pathway DNNs (DipDNN), that decomposes the nested structure to avoid the inapplicable requirement of splitting input/output equally, while ensures strict invertibility and minimum computational redundancy without hurting the universal approximation.

In general, the problem this article focuses on,  the inconsistency in bi-directional inferences, is important. Due to the inherent irreversibility of DNNs, the problem is quite challenging; the paper presents a well-designed invertible DNN architecture with rigorous theoretical analysis. However, since the datasets in the experiments are relatively small, as well as the model architecture; the scalability of the proposed method is not well illustrated.

**Strengths:**

Overall, this paper is well structured, and the idea has been explained clearly.

This paper focuses on a problem that has important practical implications but lacks effective approaches in the deep learning field. It designs a novel methodology that maintains the consistent and analytical forms of inverse solutions with the nested DNNs while reducing the computational expense and relaxing the restrictions on invertible architecture.

The paper presents a rigorous theoretical analysis to prove the strict invertibility of the DipDNN model without hurting the universal approximation. This theoretical grounding provides a strong foundation for the model's validity.

The paper conducts numerical experiments on several practical applications, including image construction and face completion, power system (PS) state estimation, etc., which show that DipDNN can recover the input exactly and quickly in diverse systems.

**Weaknesses:**

Although the proposed method appears to be significantly superior to the methods compared, I found that the baseline methods seem somehow outdated (the latest one, i-ResNet, was proposed in 2019).

The datasets in the experiments are pretty small; the DNN is also simple. For image datasets, I believe that providing some results on CIFAR10/100 (ImageNet may be impossible) and comparing them with i-ResNet can better illustrate the effectiveness of the proposed method.

**Questions:**

1. How large can this model structure scale to, and how will DipDNN with deeper and wider layers perform on complex data sets such as CIFAR?

2. As for experiments, are all the DNN architectures the same through different tasks, i.e., the DNNs on Image Construction and Face Completion and System Identification-based State Estimation?

3. In which part of the experiment the effectiveness of the proposed parallel structure for physical regularization over DipDNN are tested? Is the model performance sensitive to the hyperparameters, i.e., λ_Phy and λ_DipDNN?

4. As for the parallel structure for physical regularization over DipDNN, for physical systems with unknown priors, is the model such as equation learner jointly trained together with DipDNN?

Maybe a typo: the line below the Figure3: f(x) = λ_Phy f1(x) + λ_DipDNN f2(x), should be  f(x) = λ_DiPDNN f1(x) + λ_Phy f2(x)?

---

> ### Author Response · Authors · 2023-11-21
> **Response to reviewer uKYL:**
>
> First, we would like to thank the reviewer for the positive review. We’re pleased you found our work meaningful and well-designed and our theoretical analysis solid. The suggestions are very valuable, and we’ll use them for revision. Below, we provide responses to the questions.
>
> **Regarding baseline methods**
>
> The reviewer mentioned a very important aspect of choosing baselines to compare with. The related work is quite limited because the target problem is inverse point estimates, especially in providing analytical inverse. Most work focuses on a generative setting for inverse problems in image-related tasks, e.g., flow models. We select [1] for the classic bijective function using affine coupling. Its latest variation is [2] with applications on discriminative learning tasks, and we refer to their training details in experiments. I-Resnet [3], mentioned by the reviewer, is another typical design of invertible NN with Lipschitz regularization. To the best of our knowledge, the following work has not changed the core invertibility designs of [1] and [2] in NN architectures, which is our focus. So, we choose them as the baselines. We would really appreciate it if the reviewer had any recommendations on an up-to-date baseline method.
>
> **Considering complex dataset and model scalability**
>
> We add some experiments to test the model on CIFAR-10 and compare it with i-Resnet. We test a CIFAR-10 classification task and note that the forward-inverse mapping is built between inputs to the last feature extraction layer, where a linear classification layer is used on top of the invertible blocks. Due to the time limit, we reduce the hidden layers and control all the models to have the same number of layers (modes using MLP and having the dimension be 3072) for a fair comparison.
>
> | Model                        | i-Resnet(Conv)                                  | i-Resnet(MLP)          | NICE(MLP)          | DipDNN            |
> |------------------------------|-------------------------------------------------|------------------------|--------------------|-------------------|
> | Classification Acc. (Forward)| 74.85% (93.07% in [3] using full model)        | 69.73%                | 57.31%             | 65.23%            |
> | Running time (Forward Training)| 26.27 Sec/Epoch                                | 14.13 Sec/Epoch       | 47.93 Sec/Epoch    | 12.46 Sec/Epoch   |
> | Reconstruction Err. (Inverse)| 6.95e-5                                         | 8.32e-4               | 4.98e-3             | 3.27e-10          |
> | Running time (Inverse Testing)| 45.81 Sec                                      | 37.64 Sec             | 5.23 Sec            | 2.46 Sec          |
>
> *Clarify NN architectures:*
>
> First, we modify the baseline models for each testing case to control all the NN architectures to be as similar as possible for fair comparison. For example, NICE has a nested DNN in the affine coupling layer, and we use the same MLP for the nested DNN as our DipDNN. Then, the numbers of hidden layers differ from case to case, e.g., 3 layers for face completion task and 6-8 layers for MNIST based on cross-validation. For state estimation, it depends on the problem size, e.g., 3 layers for IEEE 8-bus network and 6-10 layers for IEEE 123-bus network based on cross-validation.
>
> *Insignts on the scalability:*
>
> For the scalability of the model, we agree with the reviewer that more practical large-scale use cases will be helpful. In existing work, the invertible model is often used for density estimation, for which image-related tasks are usually large-scale. Our work focuses on general inverse problems in physical systems, for which the scale varies from case to case. Unlike generative learning in density estimation, the inverse problem in physics targets the point estimates and usually has a critical requirement for accuracy. Thus, the emphasis of this paper is to obtain an accurate inverse solution. The current model has been tested on small synthetic examples (2-9 variables) cases as large as a 123-bus power system for physical system state estimation and MNIST data set (28$\times$28 = 784) for image construction. During our experiments, we have some observations regarding scalability. In inverse problems, a typical scalability challenge is that, with a larger problem size and larger model, the numerical errors will aggregate and lead to numerical non-invertibility. We show partial results in Figure 4 for the error propagation with respect to increasing problem dimension and nonlinearity and increasing model depth. The errors increase on an exponential scale. For this challenge, we analyze with details in Appendix A.4. While large-scale problems may aggregate errors, we can regularize bi-Lipschitz continuity to enhance inverse stability.

---

> > ### Author Response · Authors · 2023-11-21
> > **Response to reviewer uKYL (continued):**
> >
> > **Effectiveness of physical regularization**
> >
> > The physical regularization is proposed to improve the generalizability of DipDNN for physical system predictions. Therefore, both synthetic examples and system identification-based state estimation tasks have tested the parallel structure. For synthetic examples, as the physical function candidates are known as priors, we use the recovery rate of the parameters as a metric to show the forward performance and use the inverse prediction error in the extrapolation scenario to show the usefulness of physics embedding. As shown in Figure 5, even though we decrease the quantity of data used and increase the variance of data to out-of-range scenarios, the performance is stable. The second row of Figure 8 shows a similar pattern. That indicates the physics embedding in the forward can guide the prediction in the inverse process.
> >
> > *Sensitivity of hyper-parameters:*
> >
> > The reviewer pointed out a very important factor in the proposed physical regularization. During experiments, we found that sensitivity depends on 1) the observability of the physical systems and 2) the knowledge of the system priors. As this is not the focus of this work, we have some fair assumptions on both aspects based on the normal scenarios used by physics-informed learning. Moreover, we set high resolution for the hyperparameter candidates, e.g., 0.1, 0.2, …, 1.0, such that the sensitivity is relatively low in experiments. However, this is an interesting direction, and we are currently working on more general scenarios for the next work. We would appreciate it if the reviewer has any suggestions for using physics embedding in the forward to regularize the inverse counterpart.
> >
> > *Joint training:*
> >
> > The physics embedding is jointly trained with the invertible model using the same data. We trained them in turn. Specifically, during training, we found that if we train the equation learner first and then start to train both models in turn, the performance is better, with faster convergence and less likelihood of trapping into a local optima.
> >
> > *Typo:*
> >
> > Thanks for letting us know about the typo in the math expression! We’ll change it to be consistent with Figure 3.
> >
> >
> > *References*
> >
> >
> > [1] Dinh et. al. “NICE: Non-linear Independent Components Estimation.” ICLR 2015.
> >
> > [2] Ardizzone et. al. “Analyzing Inverse Problems with Invertible Neural Networks.” ICLR 2019.
> >
> > [3] Behrmann et. al. “Invertible residual network.” ICML 2019.
> >
> > Please let us know if our response clarifies your comments. If you have follow-up questions or comments, we are more than happy to discuss them with you!

---

### Official Review · Reviewer_RyQp · 2023-11-02

**Soundness:** 3 good
**Presentation:** 2 fair
**Contribution:** 2 fair
**Rating:** 5
**Confidence:** 4

**Summary:**

The authors study the problem of designing deep neural network (DNN) strctures that is invertible. As a main contribution, the decomposed-invertible-pathway DNN structure is designed to mitigate the computational redundancy of existing invertible DNN designs. Besides, a parallel DNN structure is introduced to add regularization to the invertible DNN to improve the prediction accuracy. Simulations on over a number of test cases, inlcuding the image processing and state estimation in power systems, are conducted to show the effectiveness of the proposed DNN design.

**Strengths:**

1. The problem investigated in the paper is important and interesting.
2. The idea of paper for contructing an invertible DNN is novel.

**Weaknesses:**

1. The paper is not easy to follow. See the comments below.

2. The contribution of the paper is not clear.

3. The theoretical analysis in the paper is not sufficient. See the comments below.

**Questions:**

1. The paper is hard to follow. The authors are suggested to re-organize the contents and polish the expressions in order to make it easier to read.

2. The contribution of the paper is vague. The authors are suggested to explain the advantage of the proposed approach as compared to state-of-the-art invertible DNN designs clearly. If the contribution of the proposed approach is having lower complexity, a run-time complexity analysis should be given in the paper.

3. In paragraph 3 of Sec. 3.1, the authors mention "Although the nonlinear DNN is nested in the middle, some interconnections among variables are eliminated due to the separated input/output groups, for which the comparison with regular NN is in Appendix A.1.." This is not easy to follow. The authors are suggested to explain why some interconnections among variables are eliminated in a more clear way.

4. It is not clear why fixed spliting the input and output is a important disadvantage of eisting invertible DNN designs. The authors are suggested to give some illustration and toy examples to show it.

---

> ### Author Response · Authors · 2023-11-21
> **Response to reviewer RyQp:**
>
> Thank you for taking the time to review and provide valuable comments on our work. We’re pleased to see that you found our work novel, and we appreciate your suggestions on the presentation of the paper's contents and contributions. We’ll include them in our revision. Below, please find our response to the questions.
>
>
>
> **Clarify the contributions of our work.**
>
>
>
>
> To clarify the paper's contributions, we will divide the explanation into the drawbacks of existing methods and designs to address them.
>
> *General motivations from existing work:*
>
> The neural network is not invertible in nature because the multi-layers of interconnections break a strict one-to-one correspondence. One group of previous methods approximates the numerical inverse, i.e., minimizing reconstruction error. It is difficult to ensure an accurate inverse, especially for extrapolation. For analytical inverse, several methods try to embed invertibility into the NNs, and we analyze two typical methods. One is i-ResNet [1], which has proved invertibility with regularized weights, but there is no analytical inverse form. Still, we’ll refer to its regularization to address numerical issues. The other is the NICE model [2], where the affine coupling in layers (and also the exponential nonlinearity designed later in [3]) is strictly bijective and has an easy-to-derive inverse form. We started with applying this model to the inverse problem in physical systems and found poor performance in that 1) the maximize likelihood loss function doesn’t apply as our task is discriminative learning and the data is not generated from a Gaussian distribution, 2) the random splitting of image pixels is not fair for variables with physical meaning as different splitting groups lead to different results, 3) each layer ensures the invertibility via linear paths and bypasses nonlinear mappings.
>
> *Illustrate the drawbacks of nested design and fixed input/output splitting:*
>
> Points 2) and 3) are related to the architecture limitation and representation power of existing invertible networks. To illustrate with an intuitive example, we would like to point the reviewer to Appendix A.1, where Page 14 includes three figures to compare the network architectures. Figure 9(a) is a basic layer of NICE. The nonlinear NN correlation is only between $x_{I_1}$ and $y_{I_2}$. $x_{I_1}$ and $y_{I_1}$, $x_{I_2}$ and $y_{I_2}$ are linearly correlated and $x_{I_2}$ and $y_{I_1}$ are not correlated. Therefore, at least three such layers are required to enable full coupling of all inputs with all outputs in Figure 9(b) (this can be inferred from Jacobian derivation on Page 13). Regular NN takes one layer for full coupling, so we compress Figure 9(b) into Figure 9(d) with an equivalent representation and compare the network connection with a regular NN. In Figure 11, we can observe that, while all the dimensions are coupled, the nonlinear mapping is separated into groups and isolated within each group coulping, i.e., from $x_{I_1}$ to $y_{I_1}$, from $x_{I_2}$ to $y_{I_2}$, from $x_{I_1}$ to $y_{I_2}$, and from $x_{I_2}$ to $y_{I_1}$. Many interconnections are eliminated, like nonlinear couplings of variables in $x_{I_1}$ and variables in $x_{I_2}$.
>
> Therefore, we see that the nested design in the NICE model ensures invertibility at the cost of $\sim 1.5\times$ computation complexity (i.e., at least three layers for full coupling) and representation power (i.e., an equivalent sparse representation of regular NN). This is why we are motivated to design an invertible model that relaxes the fixed splitting of variables and maintains regular NN’s representation power as much as possible.
>
> *Proposed designs to address the drawbacks:*
>
> In the proposed decomposed-invertible-pathway DNN (DipDNN) model, we first maintain the dense NN architecture and then enforce one-to-one correspondence by decomposing the nested layer of NICE into a basic invertible unit, as shown in the top right corner of Figure 2. The adoption of Leaky ReLU activation maintains invertibility in nonlinear correlation. In this way, we could prove the preserved approximation capability using the LU decomposition of weights and the latest theory of narrow deep networks [4].

---

> > ### Author Response · Authors · 2023-11-21
> > **Response to reviewer RyQp (Continued):**
> >
> > *Running time testing results:*
> >
> > As illustrated in 1), the NICE model is designed for image density estimation. Thus, for a fair analysis of run-time complexity, we conduct experiments on image-related tasks, including MNIST and face image completion, and compare the running time in Figures 6 (top right) and 7(b).
> >
> > Moreover, we add some new experiments using the CIFAR-10 dataset.
> >
> > We test a CIFAR-10 classification task and note that the forward-inverse mapping is built between inputs to the last feature extraction layer, where a linear classification layer is used on top of the invertible blocks. Due to the time limit, we reduce the hidden layers and control all the models to have the same number of layers (modes using MLP and having the dimension be 3072) for a fair comparison.
> > | Model                        | i-Resnet(Conv)                                  | i-Resnet(MLP)          | NICE(MLP)          | DipDNN            |
> > |------------------------------|-------------------------------------------------|------------------------|--------------------|-------------------|
> > | Classification Acc. (Forward)| 74.85% (93.07% in [3] using full model)        | 69.73%                | 57.31%             | 65.23%            |
> > | Running time (Forward Training)| 26.27 Sec/Epoch                                | 14.13 Sec/Epoch       | 47.93 Sec/Epoch    | 12.46 Sec/Epoch   |
> > | Reconstruction Err. (Inverse)| 6.95e-5                                         | 8.32e-4               | 4.98e-3             | 3.27e-10          |
> > | Running time (Inverse Testing)| 45.81 Sec                                      | 37.64 Sec             | 5.23 Sec            | 2.46 Sec          |
> >
> >
> >
> > *References*
> >
> > [1] Behrmann et al. “Invertible residual network.” ICML 2019.
> >
> > [2] Dinh et al. “NICE: Non-linear Independent Components Estimation.” ICLR 2015.
> >
> > [3] Dinh et al. “Density estimation using real nvp.” ICLR 2016.
> >
> > [4] Duan et al. “Minimum width of leaky-relu neural networks for uniform universal approximation.” ICML 2023.
> >
> > We would be happy to discuss any follow-up questions you may have!

---

### Meta-Review · Area_Chair_5uZT · 2023-12-09

**Metareview:**

The paper introduces a Deep Neural Network (DNN) architecture called decomposed-invertible-pathway DNNs (DipDNN) designed for bi-directional inference tasks such as state estimation, signal recovery, privacy preservation, and reasoning. The architecture is novel in that it decomposes the nested structure of DNNs, ensuring strict invertibility and minimum computational redundancy while preserving the universal approximation ability. The paper provides a theoretical analysis of the invertibility and conducts experiments on various applications demonstrating the model's effectiveness.

Strengths:
1) The paper addresses an important and challenging problem in the deep learning field, focusing on the consistency of bi-directional inferences in DNNs, which are inherently irreversible.
2) It proposes a novel DNN structure that maintains strict invertibility and reduces computational redundancy.
3) Theoretical analysis is provided to prove the strict invertibility of the DipDNN model, which does not compromise the universal approximation property of DNNs.
4) The paper demonstrates the practical applicability of the DipDNN through numerical experiments in areas such as image reconstruction and power system state estimation, showing that the model can accurately and efficiently recover inputs.

Weaknesses
1) Experiments are not strong enough. For example, the baseline methods used for comparison in the experiments seem outdated, with the latest one being from 2019, which may not adequately reflect the current state of the art. The experiments conducted use relatively small datasets and simple DNN structures, which raises questions about the scalability and effectiveness of the proposed method on more complex tasks and larger datasets like CIFAR10/100 or ImageNet.
2) Certain details are missing or unclear in the paper, making it challenging to fully evaluate the work.
3) The novelty of the theoretical contribution is questioned due to its heavy dependence on prior work by Duan et al., 2023, and existing proofs of universality for coupling layers.
4) The clarity and writing could be further improved. For example, the visual improvements in image reconstruction tasks are not easily discernible from the provided figures, and in some cases, other methods like i-ResNet seem to outperform DipDNN, but this is not addressed in the paper.

The majority of reviewers reject or weakly reject the paper. One weak accepts the paper with lower confidence. The current version could be significantly improved before reaching a solid acceptance.

**Justification For Why Not Higher Score:**

The majority of reviewers reject or weakly reject the paper. One weak accepts the paper with lower confidence.

**Justification For Why Not Lower Score:**

N.A.

---

### Decision · Program_Chairs · 2024-01-16

Reject